# Spontaneous Variation of Ventriculo-Atrial Interval after Tachycardia Induction: Determinants and Usefulness in the Diagnosis of Supraventricular Tachycardias with Long Ventriculoatrial Interval

**DOI:** 10.3390/jcm12020409

**Published:** 2023-01-04

**Authors:** Olga Durán-Bobin, Jesús Hernández, José Moríñigo, Manuel Sánchez-García, Loreto Bravo, Javier Fernández-Portales, Armando Oterino, Alba Cruz, Carlos González-Juanatey, Pedro L. Sánchez, Javier Jiménez-Candil

**Affiliations:** 1Cardiology Department, IBSAL—University Hospital, CIBER-CV, 37007 Salamanca, Spain; 2Cardiology Department, University Hospital “Lucus Augusti”, 27002 Lugo, Spain; 3Cardiology Department, University Hospital “Fundación Jiménez Díaz”, 28040 Madrid, Spain; 4Cardiology Department, University Hospital “San Pedro de Alcántara”, 10071 Cáceres, Spain

**Keywords:** supraventricular tachycardia, diagnosis, accessory pathway, slow pathway ablation

## Abstract

Background: Determining the mechanism of supraventricular tachycardias with prolongedP ventriculoatrial (VA) intervals is sometimes a challenge. Our objective is to analyse the determinants, time course and diagnostic accuracy (atypical atrioventricular nodal reentrant tachycardias [AVNRT] versus orthodromic reentrant tachycardias through an accessory pathway [ORT]) of spontaneous VA intervals variation in patients with narrow QRS tachycardias and prolonged VA. Methods: A total of 156 induced tachycardias were studied (44 with atypical AVNRT and 112 with ORT). Two sets of 10 measurements were performed for each patient—after tachycardia induction and one minute later. VA and VV intervals were determined. Results: The difference between the longest and the shortest VA interval (Dif-VA) correlates significantly with the diagnosis of atypical AVNRT (C coefficient = 0.95 and 0.85 after induction and at one minute, respectively; *p* < 0.001). A Dif-VA ≥ 15 ms presents a sensitivity and specificity for atypical AVNRT of 50% and 99%, respectively after induction, and of 27% and 100% one minute later. We found a robust and significant correlation between the fluctuations of VV and VA intervals in atypical AVNRTs (Coefficient Rho: 0.56 and 0.76, after induction and at one minute, respectively; *p* < 0.001 for both) but not in ORTs. Conclusions: The analysis of VA interval variability after induction and one minute later correctly discriminates atypical AVNRT from ORT in almost all cases.

## 1. Introduction

Paroxysmal supraventricular tachycardias involving the AV node remain one of the most frequently substrates to undergo catheter ablation in electrophysiology laboratories [1]. Catheter ablation is curative in the vast majority of cases, with little or no morbidity [1]. However, a correct diagnosis is a prerequisite for successful treatment. Hence, determining the mechanism of supraventricular tachycardias with prolonged ventriculoatrial (VA) intervals is sometimes a challenge, because the usual diagnostic manoeuvers based on pacing during ongoing tachycardia cannot be performed (i.e., the tachycardia is not sustained or is interrupted) or do not offer diagnostic accuracy (i.e., the tachycardia presents variability of RR intervals [2]).

Since the retrograde arm of atrioventricular nodal reentrant tachycardias (AVNRTs) and orthodromic reentrant tachycardias through an accessory pathway (ORTs) have different electrophysiological properties, the analysis of the behavior of VA intervals during induced tachycardia could provide diagnostic information. It has recently been proposed that VA interval variation in tachycardia may be useful to differentiate atypical AVNRTs and orthodromic reciprocating tachycardias [3]. We present the results of a prospective series in which we analysed the determinants, time course, and diagnostic accuracy of spontaneous VA interval variation in a prospective, non-selective series of patients with narrow QRS tachycardias and prolonged VA.

## 2. Materials and Methods

### 2.1. Study Population

The study was comprised of 156 patients (80 women and 76 men; mean age: 45 ± 20 years; 3 with significant structural heart disease) from three different centers, who fulfilled the following inclusion criteria: (a) previous documented paroxysmal tachycardias with a narrow QRS complex; (b) VA interval in septum >60 ms during induced tachycardias [4,5,6]; and (c) stability of electrograms recorded at the coronary sinus during tachycardia [7]. The subjects with atrial tachycardias and those who had undergone prior ablation procedures were excluded from the study. In contrast, tachycardias were not excluded because they showed fluctuating RR intervals, which in 13.5% of patients were equal to or greater than 30 ms (Table 1). The study was conducted in accordance with the Declaration of Helsinki and regional legislation with protection of personal data.

### 2.2. Electrophysiologic Study and Rhythm Classifications

After obtaining written informed consent, an electrophysiological study was performed on patients in a fasting and un-sedated state. Inserted through the right femoral vein, a 4-mm ablation catheter was placed at the high right atrium, 2 tetrapolar diagnostic catheters were placed at the right ventricular apex and at the level of the antero-septal tricuspid valve to record the His bundle signal, and the parahisian atrium and ventricle, and 1 tetra or decapolar catheter was placed at the coronary sinus. Bipolar signals were stored on conventional recording systems (Boston, Prucka Cardiolab, General Electric Medical Systems, Milwaukee, WI, USA; EPTracer, Cardiotek, Maastricht, The Netherlands). The signals were band-pass filtered between 30 and 500 kHz and recorded at 200 mm/s using the integrated calipers for the measurements. Under these conditions, the expected accuracy of the measurements is considered to be <5 ms [8].

Sustained supraventricular tachycardia was induced using conventional atrial and ventricular stimulation protocols. Of these, we excluded atrial tachycardia for analysis, characterized by a common VAAV response after entrainment from the right ventricular apex [9], absence of VA linking (i.e., variable AH and VA intervals), changes in H-H or V-V intervals that were preceded by changes in A-A intervals, or AV dissociation with rapid right ventricular pacing at a cycle length between 200 and 250 ms during tachycardia. Then, the exact mechanism of the tachycardia (atypical AVNRT or ORT) was determined according to electrophysiologic criteria [2,10,11,12] and results of catheter ablation. The classification of AVNRTs was made following the criteria proposed by Kastritsis et al. [6] because of its clarity, simplicity and wide acceptance [3,13].

### 2.3. Measurements and Definitions

We analyzed, in each patient, the first induced tachycardia that persisted for at least one minute. The VA interval was determined from the onset of the QRS complex to the bipolar atrial electrogram registered at the proximal coronary sinus (Figure 1). We consistently measured either the point at which the largest rapid deflection crosses the baseline or the peak of the largest deflection because they more or less correspond with the maximum dV/dt (intrisicoid deflection) of the unipolar records [8]. If the signal was fragmented (multicomponent electrogram), and no clear deflection behaved as the largest, we selected the peak of the first positive deflection of the electrogram for timing.

Two sets of VA interval measurements in tachycardia were performed in each patient—after induction and at one minute later. In both cases, the VA duration was determined in eleven intervals following the first interval which was excluded. This was done because quite often the catheter located in the coronary sinus can be unstable which reduces the accuracy of the VA interval measurement, and because there is a delay of interatrial conduction that even affects ORT. Only VA intervals with an identical QRS complex and atrial activation sequence were used. The following variables were then defined: arithmetic mean of beat-to-beat variation of VA interval (Mn-VA), maximum beat-to-beat variation of VA interval (Mx-VA) and the difference between the maximum and minimum VA (Dif-VA) (Figure 1). Additionally, the VV intervals containing each VA interval were measured and their variability was determined. The measurements were performed blinded to the electrophysiological diagnosis. In twenty randomly selected patients, we performed two VA interval measurements after induction and at one minute; in these cases the mean intra-observer variability was 2 ± 1 ms.

### 2.4. Statistical Analyses

The statistical analyses were performed using the SPSS software version 24 for Windows (SPSS Inc., Chicago, IL, USA). Normal and continuous variables were described using the mean and 95% confidence interval (CI) or standard deviation, whereas the continuous variables without normal distribution were displayed by the median and interquartile range. The categorical variables were summarized by the number of patients and percentages. The comparison of the categorical variables was performed using the Chi-square test (or Fisher’s exact test if *n* < 5). The comparison of 2 normal variables (determined by the Kolgomorov–Smirnov test) and continuous variables was carried out with Student’s *t* test. The comparison of >2 continuous variables was performed using the ANOVA test. Comparison of two variables without normal distribution was carried using the Mann–Whitney U test. Receiver operator characteristics (ROC) curves were created. Sensitivity, specificity, positive (PPV) and negative predictive values (NPV) of selected cut-off values were calculated. Correlations between continuous variables were performed by applying the Spearman’s test that calculates the Coefficient of Correlation (Rho). Multivariate analyses of a Dif-VA ≥ 15 ms (both after induction and one minute later) were performed using the logistic regression method. The variables included in the model were: age (years), use of drugs for induction, tachycardia cycle length (TCL) (ms), the difference between the longest and the shortest VV interval (ms) and diagnosis of AVNRT. A *p*-value < 0.05 was considered to be statistically significant.

## 3. Results

### 3.1. Electrophysiologic Diagnosis

The electrophysiologic study demonstrated ORT through an accessory pathway in 112 patients. The accessory pathway had a septal location in 57 individuals and a free-wall location (54 left free wall, and one right free wall) in the remaining 55. Atypical AVNRT was demonstrated in 44 patients, being a slow–slow form in 39 cases and a fast-slow form in 5. The demographic characteristics and the main electrophysiological values of the groups are detailed in Table 1.

### 3.2. Tachycardia Cycle Length (TCL): Values and Variability

After induction, the TCL was longer in the group of atypical AVNRT patients (Table 1); however, after analysing by tachycardia type, only those with fast–slow AVNRT had slower tachycardias (CL = 399 ± 82 ms). No differences were observed among the comparisons of the other groups: slow–slow AVNRT (CL = 386 ± 52 ms), ORT through septal accessory pathway (CL = 355 ± 60 ms) and ORT through free wall accessory pathway (CL = 346 ± 43 ms); *p* > 0.5 for all comparisons (ANOVA Tukey test). All types of tachycardias slowed discretely one minute after induction (Table 1).

As depicted in Table 1, we observed a certain degree of fluctuation of tachycardia cycle length after induction, which tended to reduce one minute later. At both time points, TCL variability was significantly greater in atypical AVRNTs. Moreover, we found a positive and significant correlation between the variability of TCL and the fluctuations of VA intervals (estimated both by the difference between the longest and the shortest intervals) in patients with atypical AVNRT. However, in subjects with ORT, both variables did not seem to be related (Figure 2).

### 3.3. VA Interval Immediately after Induction and One Minute Later

After the induction, the mean VA interval was 130 + 48 ms. As shown in Table 2, this value was significantly higher in patients with atypical AVNRT (245 ± 51 ms); no further differences were observed between the other groups. One minute after induction, the mean VA interval was similar to the previous one (131 ± 50 ms), again being significantly longer in patients with uncommon AVNRT (Table 2). In atypical AVNRT, a small decrease in the mean duration of VA intervals was observed from induction to one minute later (*p* = 0.04), and no significant changes were observed in the remaining tachycardias (Table 2).

### 3.4. Beat-to-Beat Variation of VA Intervals and Their Relation to the Time after Induction of Tachycardia

Among patients with AVNRT, there was a marked variability in the VA intervals after induction (Figure 3). No significant differences were found in the variability of the VA interval between slow–slow versus fast–slow atypical AVNRTs (*p* > 0.5 in all comparisons). Moreover, even though the variability of the VA interval at one minute continued to be considerable, it tended to decrease after induction (Figure 3).

However, as shown in Figure 3, in subjects with ORT, no relevant beat-to-beat changes in the duration of the VA intervals were seen, neither after induction nor at one minute later. The values of VA interval variability in ORT with septal versus free-wall accessory pathways were similar (Figure 3).

Additionally, in nine (20%) patients with atypical AVNRT and in eleven (10%) with ORT, the Dif-VA increased from induction to one minute, with the maximum increment being of 29 ms and 6 ms, respectively.

### 3.5. Comparison of VA Intervals in Individuals with ORT and AVRNT

As previously mentioned, the variability of VA intervals was significantly higher in subjects with AVRNT, both after induction and one minute later (Figure 3). The three variables analysed (MnVA, MaxVA and DifVA) positively and significantly correlated with the likelihood of a diagnosis of AVNRT. As can be seen in Table 3, the best correlation and the cut-off points with the highest diagnostic accuracy were obtained with the measurements taken immediately after induction. The difference between the longest and the shortest VA interval had a specificity for AVNRT of 99%. The sensitivity of this variable decreased from induction (50%) to one minute later (23%). Nonetheless, a Dif-VA ≥15 ms after induction excludes an ORT with 99% accuracy. Only one out of 112 patients with ORT (this one mediated by a septal accessory pathway) had a DifVA of 20 ms, which decreases one minute later to 6 ms. Also, none of the patients with ORT had a Dif-VA ≥ 15 ms one minute after induction. Figure 4.

### 3.6. Determinants of VA Interval Variability

Based on the results of the multivariate analysis, the diagnosis of atypical AVNRT was found to be an independent predictor of a Dif-VA ≥ 15 ms both immediately after induction (adjusted odds ratio = 56; *p* < 0.001), and one minute later (adjusted odds ratio = 18; *p* < 0.02).

## 4. Discussion

### 4.1. Main Findings

In this large series of patients with supraventricular tachycardia with AV node involvement and prolonged VA interval, a Dif-VA ≥ 15 ms after induction and/or one minute later allows atypical AVNRT and ORT to be correctly differentiated in almost all cases. This diagnostic accuracy is independent of tachycardia cycle length variation which is significantly correlated with the fluctuation of the VA interval in atypical AVNRT but not in ORT.

### 4.2. Differences in VA Conduction through the AV Node versus Accessory Pathways

The AV node generates a certain delay between the input and the output activation [14]. This can be applied either for anterograde as well as for retrograde conduction [14]. The time of AV node conduction can fluctuate depending on the interaction between different properties such as AV node recovery, facilitation and fatigue, either in normal conditions or during increased autonomic tone or high regular rates [14,15]. Because of this, retrograde conduction during AVNRT, recorded by intracavitary electrograms, often shows VA interval variations and heterogeneous atrial activation [16]. Unlike the AV node, conduction over an accessory pathway usually behaves in an all-or-none conduction [16]. Consequently, significant changes in the VA interval should not be seen in ORT with a single accessory pathway.

We observed a significant VA intervals variation immediately after AVRNT induction, which attenuated after one minute. This is probably due to the progressive regularization of the tachycardia cycle length and the ability of the retrograde way of the nodal circuit to reach a steady state [15]. In this sense, we found a significant correlation between TCL fluctuation and VA interval variability in AVNRT, which is not seen in ORT. According to our data, a Dif-VA ≥ 15 ms after induction almost completely excludes an ORT. Only one out of 112 ORTs (<1%) mediated by a septal accessory pathway showed a Dif-VA after induction of 20 ms, which after one minute was reduced to 6 ms.

### 4.3. Comparison with Previous Studies

Previous studies have shown a certain degree of spatial heterogeneity of retrograde atrial activation during AVNRT [17,18] and of conduction times (determined by the duration of the VA interval), especially after induction or when sudden changes in heart rate have occurred [19,20]. Recently, it has been proposed that a maximum VA interval variation of 10 ms or more immediately after tachycardia induction has a diagnostic accuracy of 100% for AVRNT, ruling out ORT [3]. In our opinion, the discrepancies between the two studies are a consequence of differences in the method and timing of VA interval measurements, and of the sample size, with more ORTs analyzed in our series. According to our data, the cuttoff proposed by Hadid et al. [3] is not specific enough, since 6% of all ORTs present a Dif-VA >10 ms in the 10 beats following induction (Figure 3). The greater number of ORTs analysed and the greater number of VA interval measurements performed by our group (10 versus an average of 2), as well as the unpredictable nature of VA interval variation [21] (in 10% of ORTs the Dif-VA increases from induction to one minute later) justifies this difference. We ruled out for the analysis the first VA interval after induction of tachycardia because the abrupt change in heart rate can produce some degree of intra-atrial conduction delay that is reflected in VA and even in ORTs [16]. Using this methodology, we observed that a Dif-VA of 15 ms or more after induction presented a sensitivity for the diagnosis of AVRNT of 50%.

We also believe that a second measurement one minute after induction offers a diagnostic advantage. Theoretically, the initial VA interval variation in an ORT could be due to a delay in intra-atrial conduction or to the presence of retrograde conduction through multiple accessory pathways with different conduction velocities [16]. Performing a second measurement of VA intervals later after induction allows elucidate the mechanism—when multiples accessory pathways are present the variability of VA intervals should be maintained at the same amount.

Finally, in contrast to previous studies, we have included all cases of tachycardia even with RR interval irregularity. According to our data, VA interval variability is useful in distinguishing atypical AVRTN from ORTs even in tachycardias with cycle length irregularity. This is so because RR interval variability correlates significantly with VA interval variability in atypical AVRTN but not in ORT (Figure 2).

### 4.4. Practical Implications

Many criteria used for the differential diagnosis between atypical AVNRTs and ORTs are based on their response after entrainment. Paroxysmal supraventricular tachycardia involving an AV node frequently presents some variation in R-R intervals. It is assumed that a variation equal to or greater than 30 ms makes the use of such diagnostic pacing manoeuvres in tachycardia difficult or even impossible [2,13,22]. In our study, in which 14% of the tachycardias presented such variation in cycle length, this was not a limitation for making a diagnosis based on the spontaneous variation of VA intervals.

The behavior of both the tachycardia to atrial pacing [23], or of the VA interval to right ventricular pacing maneuvers [13], allows for an accurate diagnosis of atypical AVNRT. Since our protocol does not require pacing, a diagnosis can be made with a single catheter placed in the coronary sinus, which avoids the morbidity associated with more venous punctures and reduces the cost of the procedure [24].

### 4.5. Limitations

Since only induced tachycardias persisting for at least one minute were included in the study, our results may not apply to tachycardias of a shorter duration. Such was the study protocol, designed to obtain more comparable and homogeneous results. On the other hand, our results show some discrepancies with other recent series [3], probably due to the different methodology used in the timing of VA intervals measurements. Finally, although the variable we propose does not offer an absolute diagnostic accuracy (i.e., its sensitivity is not greater than 50%), we consider that our findings provide information that could be applied during routine clinical practice.

## 5. Conclusions

Among patients with supraventricular tachycardias with AV node involvement and prolonged VA interval, measurement of the spontaneous variation of the septal VA intervals immediately after induction and one minute later allows, in most cases, a correct differential diagnosis between atypical AVNRTs and ORTs. This diagnostic accuracy is independent of tachycardia cycle length variability, which is significantly correlated with VA interval fluctuations in atypical AVRNT but not in ORT.

## Figures and Tables

**Figure 1 jcm-12-00409-f001:**
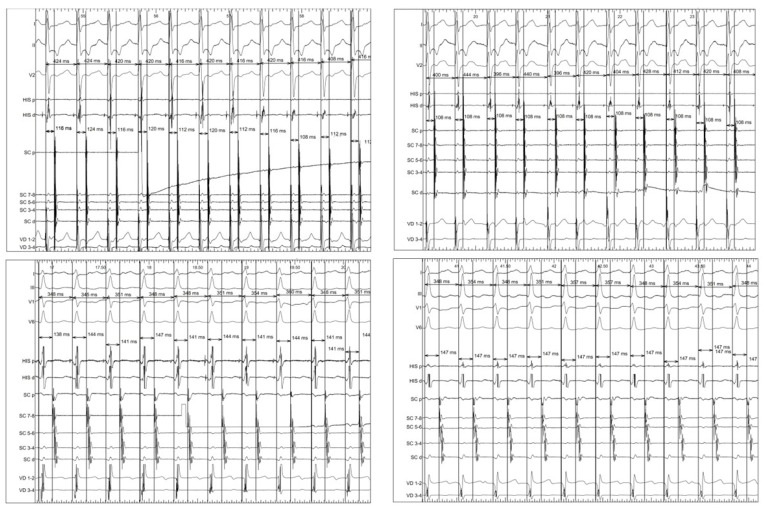
Example supraventricular tachycardias showing the measurement of variables after the induction (left) of tachycardia and one minute later (right). Top: atrioventricular nodal reentrant tachycardia. Bottom: orthodromic reciprocating tachycardia.

**Figure 2 jcm-12-00409-f002:**
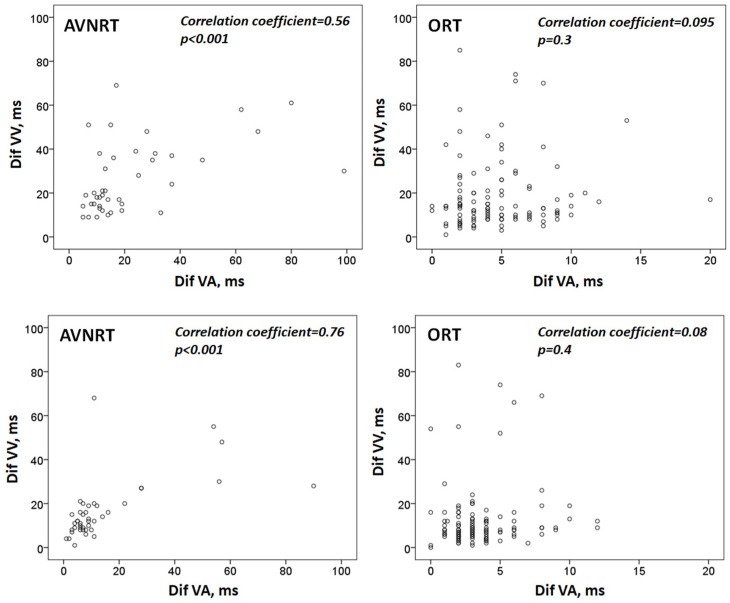
Diagram showing the correlation between tachycardia cycle length variability and VA interval variability. Top: values after induction; bottom: one minute later. Dif-VV: the difference between the longest and the shortest VV interval. Dif-VA: the difference between the longest and the shortest VA interval. AVNRT: atrioventricular nodal reentrant tachycardia. ORT: orthodromic reciprocating tachycardia.

**Figure 3 jcm-12-00409-f003:**
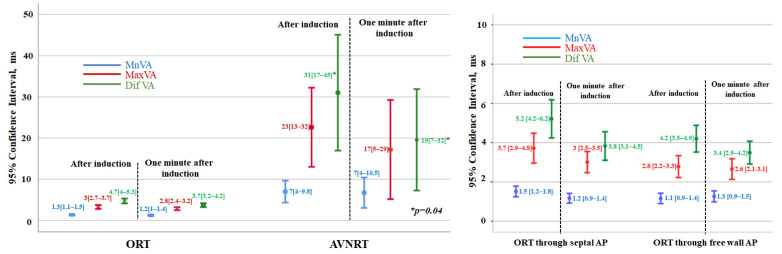
Left: Diagram showing the individual values of VA interval variation for patients with orthodromic reciprocating tachycardia (ORT) and those with atypical atrioventricular nodal reentrant tachycardia (AVNRT). In both cases, the mean beat-to-beat VA variation (Mn-VA), the maximum beat-to-beat VA variation (Mx-VA), and the difference between the maximum and minimum VA interval (Dif-VA) are depicted. For each group, the values immediately after induction and one minute later are compared. Values expressed as mean [95% CI]; *p* < 0.001 for all comparisons of ORT versus AVNRT. Right: individual values of VA interval variation for patients with orthodromic reentrant tachycardia using septal accessory pathways (AP) and free wall AP. In both cases, the mean beat-to-beat VA variation (Mn-VA), the maximum beat-to-beat VA variation (Mx-VA), and the difference between the maximum and minimum VA interval (Dif-VA) are depicted. The values obtained immediately and one minute later after tachycardia induction are shown. Values expressed as mean [95% CI]; *p* > 0.05 for all comparisons between free wall versus septal accessory pathways.

**Figure 4 jcm-12-00409-f004:**
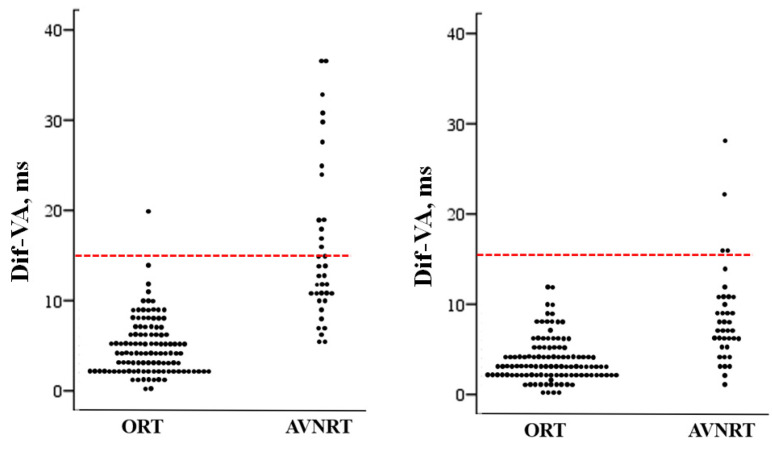
Scatterplot showing the individual values of the difference between the maximum and minimum VA interval (Dif-VA) measured after induction in patients with orthodromic reciprocating tachycardia (ORT) and those with atrioventricular nodal reentrant tachycardia (AVNRT). The line drawn at 15 ms represents the discriminant value distinguishing the patients with AVNRT from those with ORT. Left, after induction; right, one minute after induction.

**Table 1 jcm-12-00409-t001:** Description of the Study Population.

Variable	Atypical AVNRT *n* = 44	ORT *n* = 112	Statistical Analysis
Age, years	54 ± 20	41 ± 19	95% CI of the difference (5; 20); *p* = 0.001
Female sex	21 (50%)	56 (50%)	OR = 1 (95% CI: 0.5–1) *p* = 1
Drugs used in induction (atropine and/or isoprenaline)	10 (23%)	12 (11%)	OR = 1 (95% CI: 0.9–2.1) *p* = 0.05
HV interval after induction, ms	44 ± 5	44 ± 5	95% CI of the difference: [−2; 5] *p* = 0.5
TCL after induction, ms	398 ± 79	351 ± 54	95% CI of the difference: [21; 73] *p* = 0.001
Beat to beat fluctuation of CL ≥ 30 ms after induction	10 (23%)	11 (10%)	OR = 2.7 (95% CI: 1.1–7) *p* = 0.034
TCL one minute after induction, ms	402 ± 66	356 ± 42	95% CI of the difference: [19; 65] *p* = 0.001
Beat to beat fluctuation of CL ≥ 30 ms one minute after induction	5 (11%)	6 (5%)	OR = 2.2 (95% CI: 0.6–7.8) *p* = 0.1
Difference between the maximum and the minimum CL after induction, ms	21 (14–39)	13 (9–21)	Z = −4.1; non-parametric *p* < 0.001
Difference between the maximum and the minimum CL one minute after induction, ms	12 (8–20)	8 (6–13)	Z = −3.5; non-parametric *p* < 0.001

AVNRT: Atrioventricular Nodal Reentrant Tachycardia; ORT: Orthodromic Reentrant Tachycardia; CL: Cycle Length; TCL: Tachycardia Cycle Length. OR: Odds Ratio; CI: Confidence Interval.

**Table 2 jcm-12-00409-t002:** VA interval according to the type of tachycardia and the timing of its measurement.

Type of Tachycardia	VA Immediately after Induction, ms *	VA One Minute Post-Induction, ms **
Slow–slow atypical AVNRT	130 ± 54	129 ± 60
Fast–slow atypical AVNRT	245 ± 51	238 ± 55
ORT with septal accessory pathway	127 ± 42	127 ± 45
ORT with free wall accessory pathway	125 ± 36	127 ± 35

* *p* < 0.001 fast–slow AVNRT vs. others (ANOVA Tuckey’s test); ** *p* < 0.001 for fast–slow AVNRT vs. others (ANOVA Turkey’s test). AVNRT: Atrio-ventricular nodal reentrant tachycardia. ORT: Orthodromic reciprocating tachycardia.

**Table 3 jcm-12-00409-t003:** Value of Selected Variables for the Diagnosis of Atypical AVNRT.

	Sensitivity	Specificity	PPV	NPV	Area under ROC Curve *
Mn-VA ≥ 5 ms after induction	66% (52–77%)	99% (97–100%)	93% (89–97%)	78% (69–89%)	0.93 (0.89–0.97)
Max-VA ≥ 10 ms after induction	56% (43–69%)	92% (88–96%)	89% (83–95%)	85% (78–93%)	0.95 (0.91–0.98)
Dif-VA ≥ 15 ms after induction	50% (39–60%)	99% (98–100%)	96% (93–99%)	85% (79–92%)	0.95 (0.91–0.98)
Mn-VA ≥ 5 ms at one minute	23% (11–34%)	99% (98–100%)	83% (76–89%)	76% (67–87%)	0.83 (0.76–0.90)
Max-VA ≥ 10 ms at one minute	27% (15–40%)	99% (98–100%)	86% (81–93%)	76% (68–87%)	0.86 (0.80–0.93)
Dif-VA ≥ 15 ms at one minute	23% (12–34%)	100%	91% (87–95%)	77% (70–89%)	0.85 (0.78–0.92)

* *p* < 0.001 in all cases; 95% Confidence Interval in parentheses. ORT: orthodromic reciprocating tachycardia. AVNRT: Atrio-ventricular nodal reentrant tachycardia. PPV: Positive predictive value. NPV: Negative predictive value. Mn-VA: Mean of beat-to-beat variation of VA interval. Max-VA: Maximum difference of beat-to-beat variation of VA interval. Dif-VA: Difference between the longest and shortest VA interval.

## Data Availability

Not applicable.

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
