# Peer review of "Spontaneous Variation of Ventriculo-Atrial Interval after Tachycardia Induction: Determinants and Usefulness in the Diagnosis of Supraventricular Tachycardias with Long Ventriculoatrial Interval"

_jcm, 2023, doi:10.3390/jcm12020409_

Round 1

Reviewer 1 Report

Dear authors

This study is really interesting and well presented. I have some observation to make.

The references used aren't recent as only 4 of 22 are from the last decade. Please make changes in order to include more recent evidence.

In line 63 add also : " ..and regional legislative with protection of personal data.

Line 88: Please explain the morale why these criteria were chosen.

The methods and results parts are quite well presented with good figures and tables.

In lines 283-287 : the differences between the two studies are enormous. You have to explain why more extensively as crucial information are missing and you should avoid disinformation.

Line 312 : This was the finding of your study but the groups were divided initially to criteria mentioned in the methods sector. Please be more specific.

Limitations: This part needs to be change in total. You declare that your study has no limitation? In addition to this you declare that these findings are easily generalizable while other studies have opposite or different conclusions? ThisPlease rewrite this part from the beginning and present any concerning about your study 

Rephrase this sentence in order to present more clearly your conclusion.

Reviewer 2 Report

The authors should comment on the differences between this contribution and the previous report by the same authors:

Calvo D, Pérez D, Rubín J, García D, Ávila P, Javier García-Fernández F, Pachón M, Bravo L, Hernández J, Miracle ÁL, Valverde I, Gozalez-Vasserot M, Árias MÁ, Jimenez-Candíl J, Morís C. Delta of the local ventriculo-atrial intervals at the septal location to differentiate tachycardia using septal accessory pathways from atypical atrioventricular nodal re-entry. Europace. 2018 Oct 1;20(10):1638-1646. doi: 10.1093/europace/eux368. PMID: 29300867.

The authors should also comment and quote the following article:

Gilge JL, Bagga S, Ahmed AS, Clark BA, Patel PJ, Prystowsky EN, Olson JA, Steinberg LA, Padanilam BJ. Mechanism and interpretation of two-for-one response to premature atrial complexes during atrioventricular node re-entry tachycardia. Europace. 2021 Apr 6;23(4):634-639. doi: 10.1093/europace/euaa283. PMID: 33176356. 
